# Postbiotics as Metabolites and Their Biotherapeutic Potential

**DOI:** 10.3390/ijms25105441

**Published:** 2024-05-16

**Authors:** Emília Hijová

**Affiliations:** Center of Clinical and Preclinical Research MEDIPARK, Faculty of Medicine, Pavol Jozef Šafárik University in Košice, Trieda SNP 1, 040 11 Košice, Slovakia; emilia.hijova@upjs.sk

**Keywords:** postbiotics, metabolic disorders, gut microbiota

## Abstract

This review highlights the role of postbiotics, which may provide an underappreciated avenue doe promising therapeutic alternatives. The discovery of natural compounds obtained from microorganisms needs to be investigated in the future in terms of their effects on various metabolic disorders and molecular pathways, as well as modulation of the immune system and intestinal microbiota in children and adults. However, further studies and efforts are needed to evaluate and describe new postbiotics. This review provides available knowledge that may assist future research in identifying new postbiotics and uncovering additional mechanisms to combat metabolic diseases.

## 1. Introduction

All microorganisms that inhabit the human body and their genetic complement form the human microbiome [1]. The microbiome plays an important role in human health, as it supports protective, metabolic, and immune functions for optimal functioning [2]. An imbalance produces metabolites and toxins that are involved in intestinal and non-intestinal diseases, including chronic digestive disorders, chronic inflammatory disorders, autoimmunity, allergies, and metabolic syndromes [3,4]. The relationship between the intestinal microbiome and intestinal epithelial cells influences mucosal and systemic immunity, neuroendocrine function, and intestinal and non-intestinal health from childhood [5,6].

The beneficial effects of intestinal bacterial cells, including modification of the intestinal microbiota, competitive adhesion to the mucosa and epithelium, improvement of the barrier function of the epithelial lining, and modulation of the immune system, clearly depend on the state of bacteria viability [7].

Probiotics, living microorganisms with bioactive properties, have therapeutic potential (biotherapeutics) for various immune, neurological, and physiological pathologies [2,8].

Probiotics are live bacteria, and their effective amount is administered to the host, but at the end of their shelf life, they will also include potentially large amounts of dead and injured microorganisms. Little attention has been paid to the impact of non-viable bacterial cells and their components on probiotic functionality and host health.

## 2. Postbiotics

Recently, there has been increased interest in the use of alternative biotherapeutic products containing inanimate (non-living and non-viable) bacteria, microorganism-derived cell components, and metabolites that may have advantages over probiotics and are safer for use in vulnerable populations (neonates and young children, immunocompromised or critically ill patients, and those with intestinal barrier dysfunction) [9,10]. These biotherapeutic products are called postbiotics. Postbiotics is the term that has become the most widespread in recent years.

Although the exact definition of postbiotics is debated, postbiotics are defined as non-living microorganisms without losing their original function and providing a health benefit to the host [11,12,13]. Common types of postbiotics, known as metabiotics, are paraprobiotics or “ghost probiotics” and fermented infant formula. Fermented infant formula is a starter or follow-on infant formula that has been fermented with lactic acid-producing or other bacteria, and, in most cases, it does not contain viable bacteria [14,15].

The qualification of a preparation as a postbiotic includes the following: molecular characterization of the progenitor microorganisms with identification and screening of potential genes of safety concern, description of the inactivation procedure and confirmation that inactivation has occurred; detailed description of the composition of the postbiotic preparation; assessment of the safety of the postbiotics in the target host; and evidence of health benefits in the host from studies [12].

In general, inactivation of bacterial cells can be achieved by physical methods (mechanical disruption, heat treatment, γ- or UV irradiation, high hydrostatic pressure, drying, sonication) or chemical methods [16]. Heat treatment (pasteurization, tyndallization, and autoclaving) to inactivate microorganisms is widely used to ensure enzymatic and microbiological stability in the food industry, although heat treatment may not always be optimal in the production of a postbiotic preparation intended for use as a food supplement or as a food. Non-thermal inactivation techniques provide safe and stable foods with preserved overall quality and value while maintaining their sensory characteristics close to those of their fresh equivalents, and they could potentially inactivate microorganisms and generate postbiotics. Drying techniques include spray drying, air drying, and freeze drying. Spray drying creates a dry powder from a liquid or slurry by rapidly drying with a hot gas (infant formulas). Fermented foods can also contain significant amounts of non-living microbial cells, especially during long-term storage or after processing, such as pasteurization or baking. Food fermentation has a major impact on the physical properties of milk and plant-based foods and, consequently, on the potential health effects of these foods [17]. During fermentation mediated by lactic acid bacteria, various cellular structures and metabolites that affect human health can be produced, including various cell surface components, lactic acid, short-chain fatty acids (SCFAs), and bioactive peptides [18]. These effector molecules of fermented food microorganisms are thought to be similar to those produced by probiotics, but this connection has not been conclusively demonstrated. Postbiotics, while not containing live bacteria, benefit host health through mechanisms similar to those of probiotics, and this phenomenon has been referred to as the “probiotic paradox” [19].

The following postbiotics are currently available:

(1) Cell-free supernatants with biologically active metabolites produced by bacteria and yeast; (2) exopolysaccharides released from the microbiota; (3) antioxidant enzymes; (4) cell wall fragments and bacterial lipoteichoic acid (LTA); (5) short-chain fatty acids; (6) bacterial lysates from degradation of Gram-positive and Gram-negative bacteria; and (7) metabolites derived from intestinal microbiota [20].

The efficacy of postbiotics appears to depend on the interaction between the host and products generated by the microbiota (Figure 1). Postbiotics have the potential to provide new therapeutic approaches, increase the effectiveness of microorganisms, or provide a way to convert them into functional components.

## 3. Mechanism Action of Postbiotics

As complex preparations, postbiotics contain a number of bioactive compounds with multiple mechanisms of action, by which postbiotics bring benefits to human health, and is not clearly understood in most cases. The following main modes of action of postbiotics are considered [12]:
Modulation of the resident microbiota

The direct antimicrobial activity of postbiotics such as lactic acid [21] and bacteriocins [22] has been demonstrated in in vivo studies. Indirect modulation of the microbiota by postbiotics through mechanisms such as the transfer of lactic acid, which is consumed by members of the microbiota, leads to the production of SCFAs and butyrate, which have beneficial roles [23]. Postbiotics could also compete with resident microorganisms for adhesion sites if the postbiotics provide adhesins such as fimbriae [24] and lectin [25] that remain unchanged after processing. Postbiotics antagonize intestinal pathogens with metabolites and bacteriocins, prevent biofilm formation, and inactivate specific microorganisms [12,26].

b.Improvement of epithelial barrier functions

Improvement of the epithelial barrier function can be mediated by the major secreted proteins Msp1/p75 and Msp1/p40 [27,28,29] or a novel protein named HM0539 [30] of the model probiotic *L. rhamnosus* GG. Surface-associated exopolysaccharide (sEPC) of *Bifidobacterium* species (*B. longum*) can promote barrier function by reducing inflammation and preventing aberrant inflammatory responses via yet-to-be-defined signaling mechanisms [31]. Certain *Bifidobacterium* spp. induce signaling pathways, such as mitogen-activated protein kinase (MAPK) and protein kinase B (Akt), which promote tight junction function through autophagy and calcium signaling pathways [32]. Short-chain fatty acids can potentially modulate the epithelial barrier function and protect against lipopolysaccharide-induced disruption [33].

c.Modulation of local and systemic immune responses

Immunomodulatory activity is exerted by microorganism-associated molecular patterns (MAMPs) that interact with specific immune cell pattern recognition receptors (PRRs), such as toll-like receptors (TLRs) [34], nucleotide-binding oligomerization domain (NOD) receptors [35], and C-type lectins, which leads to the production of various cytokines and immune modulators at local and systemic levels [36]. These MAMPs can also occur in postbiotics if they are not destroyed or modified. Systemic immune responses are modulated by microbial metabolites that are present in postbiotic formulations, such as lactic acid [37], indole derivates of tryptophane [38], histamine [39], and branched-chain fatty acids and SCFAs [40].

d.Influencing of systemic metabolic responses

Systemic metabolic responses are influenced by metabolites or enzymes inside and on the surface of inactivated postbiotic microorganisms. Gut microbial enzymes contribute significantly to bile acid metabolism through reactions that produce unconjugated bile acids and secondary bile acids. Bile salt hydrolase (BSH) and bile acid-inducible enzymes (BAI) are important for bile acid homeostasis and contribute to the health of the host gut microbiome. Bile acids affect glucose, lipid, xenobiotic, and energy metabolism in the host [41]. Extracellular forms of BSH were observed in the probiotic *Lactobacillus johnsoni* [42]. Mullish et al. [43] showed that the restoring function of intestinal BSH contributes to the efficacy of fecal microbiota transplantation in the treatment of recurrent *Clostridioides difficile* infection.

The most studied postbiotics, SCFAs, metabolites derived from intestinal bacteria, are known modulators of host metabolism, have been suggested as potential modulating and/or preventive factors of cardiovascular diseases, including type 2 diabetes mellitus and obesity (Figure 2). Acetate, the most abundant short-chain fatty acid, favorably affects host energy and substrate metabolism through the secretion of the gut hormones glucagon-like peptide-1 (GLP-1) and peptide YY (PYY), thereby influencing appetite, reducing lipolysis and systemic levels of pro-inflammatory cytokines, and increasing energy expenditure and fat oxidation [44]. Propionate improves insulin sensitivity and glucose tolerance and modifies lipid metabolism [45], while butyrate upregulates glutathione and can favorably influence oxidative stress in the colon of healthy humans [46]. SCFAs are important for intestinal health not only as signaling molecules, but once they enter the systemic circulation, they directly affect the metabolism of peripheral tissues. They can favorably modulate adipose tissue, energy metabolism, skeletal muscle tissue, and the liver and contribute to improving glucose homeostasis and insulin sensitivity [47].

Modulation of the intestinal microbiome by postbiotics facilitates weight reduction and lowers cholesterol levels [15]. Postbiotics such as muramyl dipeptide (MDP), a component of the bacterial wall, modulate GLP-1 secretion, increase insulin sensitivity, and improve glucose tolerance [48]. Amuc_1100 (a component of the cell wall of *Akkermansia muciniphila*) and MDP, through innate immunity activation, are promising candidates for promoting cardio-metabolic benefits [49,50]. The application of live *A. muciniphila* in rodents reduced obesity, glucose intolerance, insulin resistance, steatosis, and gut permeability [50,51,52]. Pasteurization increases its effect on adiposity, insulin resistance, and glucose tolerance [53].

e.Systemic signaling through the nervous system

Postbiotics are also used in other related nervous systems. The oral administration of plasmalogens as a postbiotic, inhibits memory loss by inhibiting glial activation in mice [54] and improves cognitive function in patients with mild Alzheimer’s disease [55].

The association between depression and changes in the gut microbiome has been revealed by clinical studies, but the causal role of this bidirectional relationship of the microbiome–gut–brain axis has not yet been established. The gut microbiome is a possible source of biomarkers and a target for the treatment of depression, so postbiotics have potential use in new anti-depressant approaches [56,57].

The pathogenesis of multiple sclerosis is also associated with changes in the microbiome, reducing the number of SCFA-producing bacteria, which could be a potential target for the use of postbiotics [58,59].

Postbiotics can act in many ways, two of which postbiotics could increase their clinical benefit are modulation of the immune system through changes in the microbiome and improvement of gut barrier function by stimulation of tight junctions or stimulation of mucus production.

## 4. Postbiotics for Children Health

Postbiotics are beneficial in pediatric settings because they offer certain advantages over probiotics; their use in sensitive categories of populations such as neonates and premature children seems to be the best solution to improve the health of the microbiota in this population [60].

Fermented infant formulas and initial and follow-on infant formula that are fermented with lactic acid-producing bacteria during the fermentation process are widely available in many countries. Fermented foods are widely consumed worldwide and are estimated to make up approximately one-third of the human diet, but they are generally absent as a recommended category in dietary guidelines [61,62].

The consensus statement on fermented foods addresses how fermented foods might improve health and the status of these foods in dietary guidelines [17]. Fermented foods are defined as foods produced by required microbial species and changes in food components by enzymes. Microorganisms (autochthonous or intentionally introduced) influence the course and outcome of fermentation processes and participate in the properties of the final fermented foods. Fermented foods are beneficial due to nutritional ingredients and bioactive compounds that affect the immune system, the composition and activity of the intestinal microbiota. Fermentation is the process of producing postbiotics by microorganisms in food [63]. Food-based bioactive compounds provide a different strategy for modulating gut health compared to “probiotics” approaches, in which live microorganisms are administrated in adequate amounts.

Bioactive compounds formed by a combination of *Bifidobacterium breve* C50 (BbC50) and *Streptococcus thermophilus* ST065 showed beneficial effects on the composition and metabolic activity of the intestinal microbiota, indicating increased levels of fecal bifidobacteria, decreased counts of clostridia spores, reduced counts of *Bacteroides fragilis*, and a lower pH of stool [64,65,66]. A systematic review of five randomized controlled trials (RTCs) in which participants received a combination of *Bifidobacterium breve* C50 and *Streptococcus thermophilus* ST065 demonstrated the safety and beneficial health effects of fermented infant formula compared to standard infant formula [67]. Another review article summarized interventions with postbiotic compounds early in life, including neonates, infants, and toddlers, through adulthood, showing their mechanism effects on the microbial community, methods of application, clinical evidence, and future perspectives [14].

Bioactive compounds produced by *B. breve* C50 and *S. thermophilus* O65 in combination with prebiotics oligosaccharides (galactooligosaccharides and fructo-oligosaccharides in ratio 9:1) in infants, they significantly increased the fecal concentration of secretory immunoglobulin A (SIgA), which was the primary outcome of this study. The combination increased the number of Bifidobacteria, decreased stool, pH, and affected the overall microbial composition, with microbial colonization similar to that seen in exclusively breastfed infants. Infants fed this combined formula grew well, and the formula was well tolerated [68]. Secretory IgA plays an important role in the immunological defense of the gastrointestinal tract and serves as the first line of defense in the protection of the intestinal epithelium against toxins and pathogenic microorganisms. The formation of secretory IgA is the result of an interaction between the intestinal immune system and bacterial colonization of the intestine [69]. The intestinal epithelium in the intestinal lumen contains a layer of mucus, intestinal epithelial cells, and tight cell junctions. The outer layer of mucus is covered with many intestinal microbes, while the inner layer, which is in close contact with the intestinal epithelium, does not contain microbes. The inner layer is a protective barrier against the adhesion and invasion of bacteria. The two mucus layers, together with intestinal epithelial cells, mediate communication between intestinal and host immune cells [11].

Postbiotics show local immunomodulatory, anti-inflammatory, trophic, and antimicrobial effects. The first local effect is an increase in mucin production and, thus, an improvement in the function of the intestinal barrier. Other local effects of postbiotics include the ability to reduce inflammation, interact with lymphocytes, modulate immunity and IgA production, and promote good bacterial strains [63].

A randomized, controlled trial involving 432 healthy, full-term infants aged 0 to 28 days investigated the effects of a new fermented formula that combined specific fermented formula (FERM) called Lactofidus (Danone Nutricia, Steenvoorde, France) with galacto-oligosaccharides and fructo-oligosaccharides (scGOS/lcFOS) in a ratio of 9:1 in different groups: the first group with scGOS/lcFOS; the second group with scGOS/lcFOS + 15% FERM; the third group with scGOS/lcFOS + 50% FERM; and the fourth group with a 50% fermented formula (50% FERM) on gastrointestinal tolerance. The combination of FERM with scGOS/lcFOS was well tolerated, and a beneficial effect was observed, which indicates a lower stool pH, lower levels of *Clostridium difficile*, a lower total crying time, and higher levels of secretory IgA. Infant colitis was significantly lower in the group with scGOS/lcFOS +50% FERM than in the group with scGOS/lcFOS [70,71].

The characteristics of the postbiotic mixture depend on the strains of the fermenting bacteria. Most of the beneficial activities of probiotics are mediated by their produced postbiotics [72]. Postbiotics can thus replace probiotics by directly providing active substances that can mediate their effects on the microbiota and the immune system. Foods fermented with specific microbial strains are a natural way of providing postbiotics. Newborns received a standard formula fermented with *Lactobacillus paracasei* CBA L74 until the third month of age, which promoted the production of secretory IgA, reduced microbial diversity, and demonstrated a differentiation of fecal metabolites compared to breastfed infants. This study shows that *L. paracasei* CBA L74 is safe and well tolerated and promotes immune system maturation, microbiota, and metabolome changes. Fermented-fed infants approached the acquired immunity (according to SIgA level), and their gut microbiota and fecal profiles were similar to those of breastfed infants [73]. Secretory IgA is associated not only with acquired immunity but also with the regulation of intestinal homoeostasis and microbiota composition; the fermented formula can also improve intestinal homoeostasis and reduce the risk of developing infections [74]. Clinical trial results in young children who received the fermented formula showed that they were more protected against infectious diseases, and supplementation with heat-inactivated *L. paracasei* CBA L74 reduced the number of episodes of diarrhea and the number of cases of pharyngitis and laryngitis [75,76]. The INNOVA study presents the results of a clinical study aimed at evaluating the effect of the new infant formula Nutribén^®^ Innova 1 (Alter Farmacia S.A., Madrid, Spain) with heat-inactivated postbiotic *Bifidobacterium animalis* subsp. lactis, BPL1TM HT, with a lower protein content and increased content of docosahexaenoic acid/arachidonic acid compared to the standard formula. This new formula was considered safe as weight gain and body composition were within the normal limits according to WHO standards. Infants receiving this formula had a significantly lower prevalence of atopic dermatitis, bronchitis, and bronchiolitis [77].

The mechanisms by which fermented formulas may exert their activity remains unknown. Based on the results of in vivo studies and animal studies, possible mechanisms have been proposed. He assumes that the improved protein digestibility of the fermented infant formula could contribute to its effect. Fermented diets indicated significantly higher digestibility of crude protein in the ileum of piglets compared to a standard diet [78]. The fermentation process acidifies the product, which can improve protein digestibility through increased gastric pepsin activity. Decreased secretion of endogenous proteins reduces exposure of the gastrointestinal tract to undigested proteins, including proteases. Reduced production of microbial proteases from undigested protein may be related to the beneficial effects of fermented foods. Further studies are needed to determine the mechanism(s) by which fermented formulas may act.

Children under the age of five are particularly susceptible to infection, and evidence for the impact of postbiotics on the prevention and treatment of common infectious diseases in children under 5 years of age was summarized by Malagón-Rojas et al. [79]. Supplementation with heat-killed *Lactobacillus acidophilus* LB shortened the duration of diarrhea in therapeutic studies [80,81,82,83].

## 5. Postbiotics for Adult Health with Metabolic Disorders

Among the most common diseases in modern medicine are metabolic disorders. When normal physiological metabolic processes are disturbed due to an inappropriate diet, sedentary lifestyle, and lack of physical activity, metabolic disorders develop [84]. The most common metabolic disorders are obesity, diabetes mellitus, dyslipidemia, osteoporosis, and metabolic syndrome, which represent a real public health problem due to their worldwide increasing prevalence [85].

Metabolic syndrome (MS) is defined as the coexistence of interrelated biochemical, physiological, metabolic disturbances, and clinical factors, including central obesity, hypertension, hyperglycemia, dyslipidemia, high triglycerides, low high-density lipoprotein (HDL), and cholesterol levels that increase the risk of diabetes mellitus type 2 diabetes mellitus (DMT2) [86]. The individual components of metabolic syndrome represent an independent risk factor for cardiovascular disease, and the combination of these risk factors increases the severity and spectrum of cardiovascular disease and its subsequent complications [87]. Chronic inflammation, insulin resistance, and neuro-hormonal activation appear to be co-players in the onset, development, and transition of components of metabolic syndrome to cardiovascular disease, among all putative pathways [88].

The pathogenesis of metabolic disorders Is influenced by the composition of the intestinal microbiota. Some studies associate the development of metabolic diseases with an unbalanced or altered intestinal microbiota and bacterial diversity [89,90,91]. Postbiotics are gaining popularity because they have a beneficial effect on the intestinal microbiota; they include a number of compounds, including peptidoglycans, exopolysaccharides (EPS), teichoic acids (TA), cell wall components, cell-free supernatant, secreted proteins/peptides, organic acids, vitamins, short-chain fatty acids, and others [92,93,94,95,96].

The beneficial effects of postbiotics are reported in the literature, such as anti-inflammatory, antibacterial, immunomodulatory, anti-obesogenic, and strengthening of the immune system through multiple mechanisms, including regulation of lipid metabolism, regulation of intestinal dysbiosis, anti-carcinogenic, anti-hypertensive, and antioxidant activities [63,97,98,99,100].

### 5.1. Cell Wall Components

Cell wall components have beneficial effects on human health and are suitable for therapeutic purposes. The major components of the cell wall are peptidoglycan (PGN, also called murein, mucopeptide, or mucocomplex) and teichoic acid, which have immunomodulatory effects [101,102]. There are two types of teichoic acid: lipoteichoic acid (LTA) and wall teichoic acid (WTA). Lipoteichoic acid is bound to the membrane via glycolipid, while wall teichoic acid in the wall forms covalent bonds with peptidoglycans [103].

Peptidoglycan produced by *Lactobacillus* is responsible for inhibiting the pro-inflammatory cytokine production, interleukin-12 (IL-12), participates in T-cell regulation, and promotes the production of interferon-gamma (IFN-γ) and tumor necrosis factor-α (TNF-α). However, peptidoglycans can also stimulate production of pro-inflammatory cytokines such as TNF-α or IL-12 and IL-12p35 mRNA expression, suggesting the pro- and anti-inflammatory properties of peptidoglycans [104].

The anti-inflammatory capacity of peptidoglycan on LPS-induced macrophages was evaluated in the RAW 264.7 cell model. Peptidoglycan from Lactobacillus acidophilus significantly reduced the level of inducible nitric oxide synthase (iNOS) and cyclooxygenase-2 (COX-2), enzymes that play an important role in the inflammatory response [105]. Peptidoglycan from *L. rhamnosus* improves the immune response in immunodeficient mice infected with Streptococcus [102].

The immunoregulatory effect of postbiotics from *L. rhamnosus* CRL 1505 in human intestinal epithelial cells (IECs) and dendritic cells (DCs) was evaluated after stimulation with lipopolysaccharides. CRL1505 decreased the expression of CD40, CD80, and CD86 in DCs and increased the production of TNF-α, IL-1β, IL-6, and IL-10. Peptidoglycan (PG1505) from *L. rhamnosus* CRL1505 modulated the TLR4-triggered immune response in human IECs and DCs [95]. Bacterial cell-wall-derived muramyl dipeptide (MDP) is postbiotic, which requires a nucleotide-binding oligomerization domain containing protein 2 (NOD2). MDP decreases adipose inflammation, hepatic insulin resistance, and glucose intolerance, and it does not affect body weight or alter microbiota composition in obese mice [49].

Lipoteichoic acid from *Bifidobacterium animalis* subsp. lactis BPL1 (CECT8145) was identified as a lipid modulator with the ability to reduce fat through the insulin-like growth factor (IGF-1) pathway, suggesting its possible therapeutic and/or preventive use in metabolic syndrome and disorders related to diabetes [106]. Lipoteichoic acid from the *L. paracasei* strain D3–5 reduced high fat levels in metabolic dysfunctions and physical and cognitive disorders and suppressed inflammation by reducing the expression of pro-inflammatory cytokines IL-6, TNF-α, and IL-1β [107]. LTA produced by the intestinal microbiota is important in preventing bacterial and viral infections by releasing antimicrobial peptides [108]. Other studies have shown the anti-inflammatory potential of lipoteichoic acid, and others confirm that LTA does not reduce inflammatory processes [109,110].

### 5.2. Polysaccharides

Polysaccharides, also known as glycans, are main components of the cell wall as lipopolysaccharides, peptidoglycans, and capsular polysaccharides (CPS) secreted as exopolysaccharides (EPS) during the growth of bacteria, which are located on the cell surface or are secreted outside [111,112]. EPS are classified according to different criteria; one of the criteria is the division into homopolysaccharides and heteropolysaccharides, composed of two or more types of polysaccharides [113,114]. EPS are used in the food industry, for example, as coagulants, emulsifiers, and stabilizers, and because they have a similar function to lactic acid bacteria, they are attracting attention for use in medicine [115].

Exopolysaccharides produced by lactic acid bacteria have biofunctional properties, such as antioxidant effects by the scavenging of free radicals, immunomodulatory effects, the regulation of intestinal microbiota, and cholesterol-lowering activity by binding to free cholesterol [116,117,118]. Exopolysaccharides from *Lactobacillus plantarum* L-14 extract inhibit adipogenesis through TLR2 and AMP-activated protein kinase (AMPK) signaling pathways, oral intake of *L. plantarum* L-14 extract ameliorates obesity, and obesity-associated diseases in vivo and can be used for the prevention or treatment of obesity and obesity-associated metabolic disorders [119]. Exopolysaccharides from *Bacillus subtilis* sp. (BSEPS) decreased fasting blood glucose, increased serum insulin levels, and improved dyslipidemia in experimentally induced diabetes [120]. Preclinical cross-sectional studies involving microbial models of EPS and models of diabetes were mentioned in a review article by Cabello-Olmo et al. [121].

### 5.3. Cell-Free Supernatants

Cell-free supernatants (CFS) obtained by the centrifugation of cell cultures contain biomolecules and metabolites with beneficial effects for maintaining human health [122,123,124]. CFS has been associated with biological activities such as reducing oxidative stress, anti-inflammatory, anti-cancer, and antibacterial effects, as well as suppressing biofilm formation, which may be useful for clinical practice. Cell-free supernatants of probiotic strains *Lactobacillus acidophilus*, *Lactobacillus casei*, *Lactococcus lactis*, *Lactobacillus reuteri*, and *Saccharomyces boulardii* show anti-inflammatory and antioxidant properties, acting first on intestinal epithelial cells and then on the immune system. The selection of the strain of probiotics used in nutraceutical preparations requires special attention because not all probiotics have the same immunomodulatory effects. Metabolites produced by these probiotics reduced the production of prostaglandin-2 (PGE-2) and IL-8 in the human intestinal HT-29 cell line. Probiotic supernatants differentially affect IL-1β, IL-6, TNF-α, and IL-10 production by human macrophages, indicating anti-inflammatory as well as dose-dependent radical scavenging activity [125].

The treatment of mice with *L. rhamnosus* GG (LGGs) supernatant exposed to a high-fat/high-fructose diet and intermittent hypoxia (HFDIH) reduced body weight gain and body fat mass and suppressed liver injury by decreasing de novo lipogenesis and increasing lipid β-oxidation. These effects relate to the regulation of the adiponectin-FGF21 (fibroblast growth factor 21) axis. HFDIH-induced metabolic dysfunction in LGG-treated mice reduced adipose tissue inflammation by decreasing pro-inflammatory cytokine levels and increasing FGF21 expression [126]. The anti-diabetic effects of live and dead probiotic strains (*L. brevis* CCFM648, *L. casei* CCFM419, *L. rhamnosus* Y37, *L. plantarum* X1, and *L. plantarum* CCFM36) were compared in mice fed a high-fat diet and with Streptozotocin-induced diabetes mellitus type 2. Multi-strain death showed a decrease in fasting blood glucose (FBG), glycosylated hemoglobin A1c (HbA1c), and leptin levels, as well as increasing GLP-1 levels. The study found that live probiotics alleviated the symptoms of hypoglycemia in the host more effectively than death multi-strain probiotics by reducing insulin resistance. This effect was associated with butyrate production by gut microbiota and the inflammatory response and is more effective for live probiotics than dead probiotics strains [127].

### 5.4. Extracellular Vesicles

Extracellular vesicles (EVs) are secreted by intestinal microbiota that release proteins, enzymes, polysaccharides, and toxins that are able to easily pass through the mucus layer and interact with the host via microbe-associated molecular pattern (MAMP) and pathogen-associated molecular pattern (PAMP), pass outside the intestine, and spread to other organs and tissues [121,128,129]. EVs have an important role in the development and progression of diabetes mellitus type 1 (DMT1) and type 2 (DMT2) [130,131].

In one study, extracellular vesicles from *Akkermansia muciniphila* improved the intestinal permeability and glucose tolerance in mice fed a high-fat diet (HFD) that induced DMT2; this effect was likely due to the regulation of tight junction proteins that prevented the risk of metabolic endotoxaemia. This study also showed that EVs improved the intestinal barrier function of Caco2 cells in in vitro experiments [132]. Extracellular vesicles from pasteurized *Akkermansia muciniphila* MucT reduced body weight, food consumption, and metabolic parameters and acted to prevent HFD-induced obesity by changing epididymal white adipose tissue (eWAT). *A. muciniphila* and EVs decreased the expression of lipoprotein lipase (LPL), transforming growth factor beta (TGF-β), IL-6, TRL-4, and TNF-α and significantly decreased the Firmicutes/Bacteroidetes ratio, and these results suggest that they could prevent HFD-induced adipo-inflammation [133]. The anti-inflammatory action of *Propionibacterium freudenreichii* EVs has been demonstrated and is essentially associated with affecting the nuclear factor kappa B (NF-κB) pathway, which is dependent on their concentration [134].

### 5.5. Short-Chain Fatty Acids

Short-chain fatty acids (SCFAs) are volatile fatty acids produced by intestinal microbes by anaerobic microbial fermentation of non-digestible polysaccharides [20]. Dietary fibers, such as undigested dietary carbohydrates from food entering the colon, are a food source for the microbiota. If consumption of fibers increases, SCFA production will also increases [135]. The most representative intestinal SCFAs are acetate, propionate, and butyrate in a ratio of 60:20:20, which promote the growth of beneficial bacteria by changing the intestinal environment [136,137]. The quantities of each SCFA are influenced by the microbiota profile, with Firmicutes primarily supporting butyrate production and Bacteroidetes primarily acetate and propionate production [138].

SCFAs contribute to the metabolism of lipids and carbohydrates, reduction of inflammatory risk disease [139], anti-obesity, and diabetes prevention properties [99,140,141], reduced appetite and fat accumulation that can help prevent cardiovascular diseases [142], and maintenance of intestinal barrier function, and they act in the process of remodeling adipose tissue and the immune system [143].

SCFA can act via two different mechanisms: by directly acting on enterocytes, maintaining gut barrier integrity, driving enterocytes, increasing mucin production, increasing tight junction proteins and to regulate peptides, and decreasing LPS and pH, or indirectly by acting on SCFA regulates the inflammatory and immune response (increase in T-cells, and anti-inflammatory ILs, and decrease in pro-inflammatory cytokines), blood pressure (binding to G protein-coupled receptor 41 (GPR-41), increase in satiety, and maintaining lipid and glucose homeostasis through mechanisms:(a)Inhibition of K/HDAC (lysine/histone deacetylase) leads to histone hyperacetylation.(b)Activation of signaling transduction by AMPK (AMP-activated protein kinase) and NFκB, and GLP-1 and PYY secretion in intestinal enteroendocrine L-cells caused by the SCFAs binding to GPR. GLP-1 and PYY enter the systematic circulation, which is beneficial in various tissues and cells.(c)Butyrate acts as a ligand of AHR (aryl hydrocarbon receptor) and PPARy (peroxisome proliferator-activated receptor γ), leading to the expression of genes that are dependent on these transcription factors [91].

SCFAs control the immune system through mechanisms including the activation of (GPRs) and the inhibition of histone deacetylase [144]. The anti-cancer and anti-inflammatory effects of extracellular SCFA metabolites of SCFAs and individual SCFAs were investigated on various cancer cells. Extracellular metabolites produced by the selected *E. coli* KUB-36 strain demonstrated a greater cytotoxic effect on MCF breast cancer cells than on colon cancer and leukemia cancer cells. The metabolites showed anti-inflammatory effects on LPS-induced macrophage cells, suppressing the production of IL-6, IL-8, IL-1β, and TNF-α cytokines [145]. SCFAs may have various anti-inflammatory activities due to the release of prostaglandins, cytokines, and chemokines from human monocytes and peripheral blood mononuclear cells [146].

Administration of inulin-propionate ester to overweight adults for 24 weeks significantly stimulated the release of PYY and GLP-1 hormones and reduced appetite and caloric intake. The long-term intake of inulin-propionate ester led to a significant decrease in weight gain and distribution of intra-abdominal adipose tissue and a decrease in intra-hepatocellular lipid content [147].

Colonic acetate infusions in overweight and obese men promoted fasting fat oxidation and peptide YY concentration, increased fasting circulating acetate, postprandial glucose, and insulin concentration, and decreased TNF-α [148]. In a similar study, dosing SCFA mixtures into the distal colon of overweight and obese men with normal glycemic levels significantly increased fasting fat oxidation and postprandial peptide YY secretion and decreased lipolysis in test subjects [149]. Other clinical studies have also shown that postbiotic therapy reduces weight gain and increases energy expenditure. SCFAs activate GPR-41 and GPR-43, promoting the secretion of the gut hormones PYY and GLP-1, which reduce food intake and appetite, ultimately causing weight loss [150,151]. SCFAs contribute to maintaining health and improving disease states, including cardiovascular diseases. Butyric acid reduced angiotensin II-induced hypertension in mice [152]. Diet-related microbial metabolites such as SCFAs have properties to reduce blood pressure, cardiac hypertrophy, and fibrosis. Metabolites have known ligands (for example, SCFA receptors such as GPR41, GPR43, GPR109a, and Olf78 in mice/OR51E2 in humans) that could be modified as therapeutic targets in the treatment of hypertension. Therefore, SCFA supplementation may have a beneficial effect on the prevention of cardiovascular diseases [153].

### 5.6. Enzymes

Enzymes such as peptidases, lipases, amylases, proteases, ureases, and antioxidants catalyze various biological processes. These are active proteins found in all living organisms. Enzymes derived from microorganisms break down components such as carbohydrates, fats, and proteins into monomers, thereby aiding human digestion and bioavailability and improving the taste and texture of food [154,155].

Lactic acid bacteria (LAB) are a subgroup of microorganisms that produce lactic acid and are undoubtedly the most important group of bacteria in the food industry. They produce a variety of bioactive compounds, including functional EPSs, enzymes, vitamins, anti-inflammatory agents, peptides, and antimicrobial products, that are used in foods and beverages to improve the safety of fermented foods and inhibit harmful bacteria [156]. LABs scavenge free radicals through antioxidant enzymes such as catalase (CAT), glutathione peroxidase (GPx), NADH-oxidase, and superoxide dismutase (SOD) and combat reactive oxygen species (ROS) and oxidative stress [20]. Reducing ROS levels using antioxidant enzymes could be useful in the prevention of certain diseases. Anti-inflammatory effects of CAT or SOD-producing *Lactobacillus casei* strain BL23 have been reported using a murine model of 2,4,6-trinitrobenzene sulfonic acid (TNBS)-induced Crohn’s disease. Treatment of Crohn’s disease mice with CAT or SOD producing *Lactobacillus casei* strain BL23 showed faster recovery of initial weight loss, increased enzymatic activities in the intestine, and less extent of intestinal inflammation compared to mice that did not receive bacterial supplementation [157]. In the animal study, *L. plantarum* 30B with the highest catalase activity and *L. acidophilus* 900 with the highest superoxide dismutase-like activity were used. The results of the study showed that Lactobacillus strains with dismutase-like activity were more effective in reducing intestinal inflammation than catalase-producing strains, suggesting that the breakdown of superoxide anion radicals is important in the intestinal inflammation process [158].

### 5.7. Bacteriocins

Postbiotics generated by the microbiota include peptides. Antimicrobial peptides (AMPs) are groups of effectors with the ability to inactivate microorganisms by membranolytic effects as well as by interacting with specific molecular targets through the nonlytic pathway [159]. Bacteriocins are a group of antimicrobial peptides produced by Gram-positive and Gram-negative bacteria with a broad spectrum of effects. The advantages of the four classes of bacteriocins are based on pillars such as spectrum, stability, bioengineering, diversity, production, and safety [160]. Nisin, a class I bacteriocin (lantibiotics) produced by Gram-positive bacteria belonging to the species Lactococcus and Streptococcus, is used as a biopreservative in food and is widespread in biomedical fields. Nisin inhibits the growth of drug-resistant bacterial strains such as methicillin-resistant *Staphylococcus aureus*, *Streptococcus pneumoniae*, *Enterococci,* and *Clostridium difficile*. Nisin has anti-microbial activity against both Gram-positive and Gram-negative disease-associated pathogens [161]. The bacteriocin produced by *Enterococcus faecalis* KT11 is antagonistically effective against a variety of Gram-positive and Gram-negative bacteria, including vancomycin- and/or methicillin-resistant bacteria. It has a broad antimicrobial spectrum, thermal stability and stability over a wide pH range, and can be used as a potential bio-preservative in food. Bacteriocin KT11 alone or in combination with conventional antibiotics represents an option for use in the treatment of multidrug-resistant clinical pathogens [162].

Plantaricin EF (PlnEF), a bacteriocin class II, produced by *Lactobacillus plantarum,* exhibits anti-inflammatory activity against 2,4,6-trinitrobenzene sulfonic acid (TNBS)-induced inflammatory bowel disease in a mouse model. A non-significant amount of colonic TNF-α and IL-6 was observed in mice using *L. plantarum* NCIMB8826. This results suggest a role for *L. plantarum* PlnEF production in favoring digestive tract health [163]. Bacteriocin plantaricin EF from *Lactobacillus plantarum* in obese mice caused a decrease in body weight and food intake and preserved epithelial barrier integrity in vitro without changes in the composition of the microbiota [164].

### 5.8. Vitamins

Vitamins are essential nutrients that participate in various biological processes in the body. Changes in eating habits, an unbalanced diet, or malnutrition lead to a lack of vitamins, which changes the biological processes in the body [94,165]. Humans are unable to synthesize most vitamins; they must be obtained through appropriate supplementation. Vitamin K and vitamin B are synthesized by intestinal bacteria and probiotic microorganisms. Vitamins synthesized by intestinal bacteria are more beneficial to the consumer and cause fewer side effects than chemically manufactured vitamins. Water-soluble B vitamins are synthesized by the human intestinal microbiota and are absorbed in the large intestine, while B vitamins from the diet are absorbed in the small intestine [166,167]. Folate is an essential dietary component in human nutrition that occurs naturally in foods and is one of the many metabolites produced by bacteria of the genus Lactobacillus and Bifidobacterium, species *B. adolescentis* and *B. pseudocatenulatum*, which are involved in several metabolic pathways [168]. Folic acid showed a protective effect against nephropathy in diabetic rats, reduced oxidative stress by reducing malondialdehyde levels, and significantly reduced the histopathological score and percentage of apoptotic kidney cells [169]. The administration of folic acid can reduce glycemic levels and improve the activity of superoxide dismutase and catalase in experimentally induced diabetes mellitus [170].

Cobalamin (vitamin B12) is synthesized only by the microbiota and cannot be synthesized by animals, plants, or fungi [165]. Enzymes involved in cobalamin biosynthesis are encoded genes identified in probiotics within the genus Lactobacillus, like *L. reuteri*, *L. sanfranciscensis*, *L. coryniformis*, *L. plantarum*, *L. rossiae*, and *L. fermentum*. They can serve as potential alternatives in the industrial production of this vitamin [171]. Vitamin B12 directly scavenges reactive oxygen species and regulates glutathione (GSH) maintenance. Vitamin B12 and GSH deficiency is associated with several diseases related to oxidative stress, including DMT2 [172].

Riboflavin (vitamin B2) and niacin (vitamin B3) are most commonly synthesized by the human intestinal microbiota [173]. Riboflavin plays an important role in cellular processes, including mitochondrial metabolism, stress responses, the biogenesis of vitamins, and cofactors. The authors describe the function of riboflavin on overall human health and its benefits in the treatment of patients with congenital metabolism disorders [174]. Riboflavin deficiency is a significant predictor of anemia, suggesting that the correction of riboflavin deficiency may potentially play a small but important protective role [175].

### 5.9. Cell-Free Lyzates

The most common risk factor for metabolic diseases is obesity. Research on the association between obesity and intestinal microbiota identified obesogenic and anti-obesogenic bacterial species. The phylum Firmicutes is positively associated with the development of obesity, while Bacteroidetes exhibit anti-obesogenic activities. The composition and abundance of intestinal microbiota can be changed by the diet [176].

The modulation of intestinal microbiota and the anti-obesity effect were studied in a pig obesity model after supplementation with the heat-killed *Ligilactobacillus salivarius* strain 189 (HK LS 189). The intestinal microbiota of pigs changed; the proportion of Prevotella decreased; and Parabacteroides increased. The result of the analysis of the composition of the microbiota showed at the phylum level abundances (Firmicutes, Bacteroidetes, and Proteobacteria) of 97.7% and 87.2% in the control group and in the supplemented group, respectively [177]. Diet-induced obesity in mice after supplementation with heat-killed *Lactobacillus plantarum* L-137 (HK L-137) alleviated levels of glucose, cholesterol, liver enzymes such as alanine aminotransferase and aspartate transaminase, and weight gain. Administration of HK L-137 significantly reduced the plasma level of lipopolysaccharide-binding protein and suppressed the expression of inflammation-related genes in the epididymal adipose tissue, contributed to the improvement of obesity-induced metabolic abnormalities, reduced liver damage by increasing intestinal permeability, and reduced translocation of endotoxin [178].

The paraprobiotic, heat-killed lactic acid bacteria (HLAB) *L. mesenteroides* 4 (LMDH04) and *L. kefiri* DH5 (LKDH5) with prebiotics (polyphenol-rich wine grape seed flour) significantly reduced body weight gain, visceral adiposity, and plasma triacylglycerol concentrations in mice. A synergic combination of prebiotic and paraprobiotic HLAB has been shown to have anti-obesity effects and is useful in the prevention of obesity and obesity-related diseases, especially in immunocompromised individuals [179]. The effect of heat-killed *Lactiplantibacillus plantarum* K8 (K8HK) on the anti-differentiation of 3T3-L1 preadipocytes as well as the anti-obesity effect was investigated in high-fat diet mice. K8HK showed the greatest anti-obesity effect, while live *L. plantarum* K8 showed cytotoxicity effect. K8HK reduced the synthesis of triglycerides in liver and adipose tissues by suppressing genes related to fat metabolism. The data show that heat-treated probiotic K8HK increases the expression of negative regulators, including SOCS-1 (suppressor of cytokine signaling-1), by acting on 3T3-L1 cells. SOCS-1 and possibly other negative regulators can block the JAK2-STAT3 pathway. Consequently, the levels of transcription factors such as peroxisome proliferator-activated receptor γ (PPARγ), CCAAT/enhancer binding protein α (C/EBPα), lipogenic enzymes, and fatty acid binding protein 4 (FABP4) were decreased in K8HK-treated cells. K8HK could be suitable for preventing obesity by inhibiting genes related to fat metabolism [180].

## 6. Safety and Efficacy of Postbiotics versus Potential Side Effects

Postbiotics have a beneficial effect on the host; they are compounds and/or cell wall components produced by living bacteria or released after bacterial lysis. Postbiotics do not expose the body to the risk of ingesting live bacteria and do not have a lasting effect if treatment is discontinued. Further research on the bioactivities of these metabolites to reveal the potential use of postbiotics in medicine is needed. Studies using in vitro intestinal model systems may be essential in this area. Postbiotics can treat or prevent a wide range of diseases, including those for which there are no effective etiological therapies [181]. Postbiotic dietary supplements are more stable than probiotics and prebiotics; they help with bio preservation and reduce the formation of biofilm in food thanks to their organic acids, bacteriocins, and other antibacterial activities. Organic acids lower pH and acidify the cell membranes of bacteria. Lactic acid and acetic acid could be used to create new antibacterial agents for widespread use in the food industry. Postbiotics are included in the active packing system “antimicrobial active packing”, which protects food from microbial decomposition during transportation and storage as antimicrobial agents [182]. The use of postbiotics as therapeutic agents faces several obstacles that must be addressed before they can be successfully implemented in clinical practice. The important issue is a lack of standardized postbiotic definitions and criteria, which makes it difficult to compare research and products. The next problem is optimizing the dose, which is difficult because the optimal dose is determined by factors such as the specific postbiotic, the desired disease, and the patient’s characteristics. The distribution technique must be carefully selected, taking into account characteristics such as stability, bioavailability, and targeted distribution to specific areas of the gastrointestinal tract. Understanding the mechanism of action, specificity of effects, and integration with conventional therapy are also persistent problems. Despite these obstacles, continuous research and development can pave the way for the successful use of postbiotics in therapeutic applications.

## 7. Conclusions

Foods with probiotics and prebiotics ensure that the gut has enough essential components for the production of postbiotics, which include metabolites and components produced by live bacteria that can have beneficial effects on the host by increasing the bioactivity of postbiotics. The bioactivities of postbiotics include anti-obesogenic, anti-inflammatory, anti-proliferative, antioxidant, hypocholesterolemic, and immunomodulatory effects.

Postbiotics have several advantages over probiotics, including a longer shelf life, easier storage, and lower maintenance requirements at low temperatures. Most of the studies conducted investigating the effects of postbiotics have been validated in animal models, and several clinical studies have focused mainly on the use of SCFAs and heat-killed probiotic bacteria in overweight and obese individuals. To fully understand the mechanisms of obesity, more clinical studies are needed to determine their potential value in the treatment and prevention of overweight and obesity, especially regarding dosing and safety aspects. It is likely that future clinical research will focus on the composition of postbiotics and determine their clinical efficacy and prophylactic effects, as well as their other biological activities, as more information about postbiotics becomes available. Postbiotics research will bring new knowledge and increase their use in the fields of the food industry and pharmacy. These efforts will contribute to the production of safe, natural, and pure products that will not contain any residues, thereby protecting human health. It is necessary to focus on monitoring the health effects of postbiotics and their specific mechanisms in other diseases. Postbiotics research presents a great challenge and opportunity for all those working in this field.

## Figures and Tables

**Figure 1 ijms-25-05441-f001:**
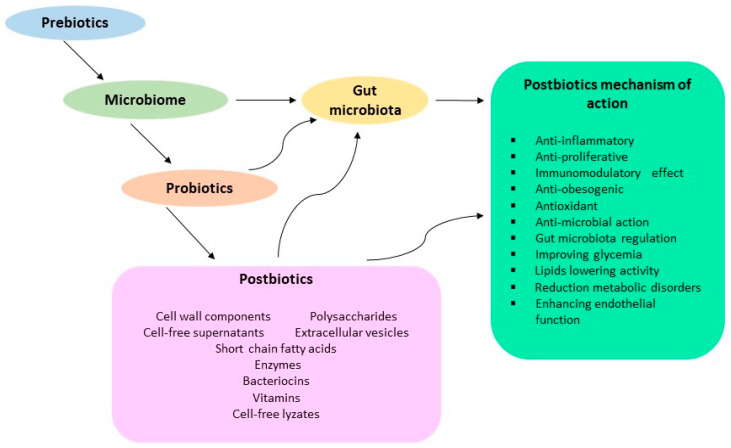
Postbiotics mechanism of action. Arrow indicate the relationship between gut microbiota, prebiotics, probiotics and different postbiotics. More details in text.

**Figure 2 ijms-25-05441-f002:**
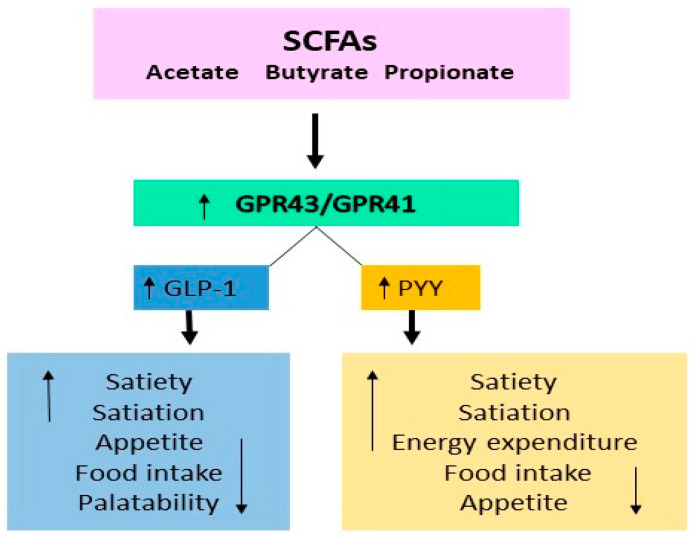
Anti-obesogenic effect of SCFAs. SCFAs activate GPR43 and GPR41 that support secretion of GLP-1 and PYY, manifesting anti-obesogenic effects. Most details in text.

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
