# Peer review of "Postbiotics as Metabolites and Their Biotherapeutic Potential"

_ijms, 2024, doi:10.3390/ijms25105441_

Round 1

Reviewer 1 Report

Comments and Suggestions for Authors

Title;  Postbiotics as metabolites and their biotherapeutic potential
Comments;  In my view, the results obtained in this study are worthy for publication. The manuscript needs major essential revision before publication. I would like to overview the revised version of the manuscript. I have the following comments/suggestions for authors to address before final decision on the manuscript.
1. The study would benefit from a clearer definition of terms such as "postbiotics," "paraprobiotics," and "metabiotics" to ensure consistency and avoid confusion among readers.
2. The manuscript mentions various studies and findings, but it lacks specific citations for many claims. Adding appropriate references will strengthen the credibility of the study and allow readers to access further information.
3. The text mentions the mechanisms of action of postbiotics, it would be beneficial to provide more detailed explanations or examples to help readers understand how postbiotics exert their effects on the host.
4. Consider integrating figures or diagrams to illustrate key concepts discussed in the text, such as the mechanisms of action of postbiotics or the interaction between postbiotics and the host microbiota.
5. Identify and discuss emerging research areas or trends related to postbiotics that warrant further investigation.
6. Ensure consistency in the nomenclature used for bacterial strains and components throughout the text. For example, use the same naming convention for strains of Lactobacillus (e.g., L. rhamnosus) and ensure consistency in abbreviations (e.g., TNF-α vs. TNF-alpha).
7. Explore molecular mechanisms underlying the observed immunomodulatory effects of cell wall components, such as signaling pathways involved in cytokine production or interactions with host receptors.
8. In the Introduction section the author should refer to the research paper and comment on recent in-silico techniques. It will be good information for the readers. I would like to recommend several papers, among many others, providing further explanation on this topic:PMID: 33749525 PMID: 23564489 PMID: 23852834
9. The author should clarify the specific metabolic disorders and molecular pathways targeted by postbiotics to provide a more focused understanding of their therapeutic potential.
10. The author should provide insights into the mechanisms underlying the observed bioactivities of postbiotics, including their effects on specific cellular pathways and physiological processes.
11. The author should discuss the potential synergistic effects of combining postbiotics with probiotics or prebiotics to enhance their therapeutic efficacy and broaden their applications.
12. The author should consider the implications of postbiotic research for personalized medicine approaches, including identifying biomarkers for predicting individual responses to postbiotic interventions.
13. The author should discuss the potential impact of postbiotics on public health and healthcare systems, including their role in preventing and managing chronic diseases and reducing healthcare costs.
14. The author should discuss the potential environmental and sustainability implications of scaling up postbiotic production and consumption, including resource use and waste management considerations.
15. Clarify the definition of postbiotics for readers who may not be familiar with the term.
15. Expand on the potential mechanisms through which postbiotics could exert their effects on metabolic disorders.
16. Provide more specific examples of natural compounds obtained from microorganisms that have shown promise as postbiotics.
17. Discuss any existing research on the safety and efficacy of postbiotics in humans, particularly in the context of metabolic disorders.
18. Consider including information on the potential side effects or limitations associated with the use of postbiotics.
19. Consider including a section on the current methods used to identify and characterize new postbiotics.

Comments on the Quality of English Language

 Minor editing of English language required

Author Response

Cover letter for reviewer 1
Thank you for the review and I attach explanations. 
Postbiotics maintan host health and many act as a mediator for the positive effects via a number of 
pathways, but the targeted mode of action is still being studied, therefore I am not including them in the 
manuscript except of mechanism action of SCFA, just explains them in individual points.
Some proposed necessary misspelling have been repaired in the manuscript.
Title; Postbiotics as metabolites and their biotherapeutic potential
Comments; In my view, the results obtained in this study are worthy for publication. The manuscript 
needs major essential revision before publication. I would like to overview the revised version of the 
manuscript. I have the following comments/suggestions for authors to address before final decision on 
the manuscript.
1. The study would benefit from a clearer definition of terms such as "postbiotics," "paraprobiotics," 
and "metabiotics" to ensure consistency and avoid confusion among readers.
As I write in manuscript „Although the exact definition of postbiotics is debated, postbiotics are defined 
as non-living microorganisms without losing their original function and providing a health benefit to the 
host [11-13]. Common types of postbiotics, known as metabiotics are paraprobiotics or “ghost probiotics” 
and fermented infant formula“ there is no exact definition, so we can also define 
postbiotics as „non-viable bacterial products or metabolic products from microorganisms that have 
biological activity in the host“, or 
postbiotics are metabolites and cell-wall components that are beneficial to the host and are released 
by living bacteria or after lysis,
parabiotics as “inactivated microbial cells (non-viable) that confer a health benefit to 
the consumer” and 
metabiotics as “ the structural components of probiotic microorganisms and/or their metabolites and/or 
signaling molecules with a determined (known) chemical structure that can optimize host-specific 
physiological functions“.
2. The manuscript mentions various studies and findings, but it lacks specific citations for many claims. 
Adding appropriate references will strengthen the credibility of the study and allow readers to access 
further information.
There is a sufficient number of 180 citations in the manuscript, each citation represents work whose 
results are described not only in this manuscript but also in other articles, so I think there is no need to 
increase the citations.
3. The text mentions the mechanisms of action of postbiotics, it would be beneficial to provide more 
detailed explanations or examples to help readers understand how postbiotics exert their effects on 
the host.
Part 3 in manuscript is focused on the mechanism of action of postbiotics. The main modes of action 
of postbiotics are considered:
- Modulation of the resident microbiota
- Improvement of epithelial barrier functions
- Modulation of local and systemic immune responses
- Influencing of systemic metabolic responses
- Systemic signaling through the nervous system
The description of individual mechanisms is included and described for individual postbiotics. Research 
and development in the field of postbiotics continues, while the exact mechanism of some postbiotics is 
still not explained.
4. Consider integrating figures or diagrams to illustrate key concepts discussed in the text, such as the 
mechanisms of action of postbiotics or the interaction between postbiotics and the host microbiota.
In manuscript are two figures: Figure 1. Postbiotics mechanism of action, which presents the 
relationships between probiotics, prebiotics, and postbiotics and mechanisms of action of postbiotics. 
Figure 2. Anti-obesogenic effect of SCFAs show as SCFAs activate GPR43 and GPR41 that support 
secretion of GLP-1 and PYY, manifesting anti-obesogenic effects. 
These relationships are explained in the text of the manuscript. The figures present simple schemes of 
the effects, because their mechanisms are constantly being studied
5. Identify and discuss emerging research areas or trends related to postbiotics that warrant further 
investigation.
Research and development in the field of postbiotics continues primarily in the food industry and the 
pharmaceutical industry, the development of new techniques will certainly contribute to the identification 
of new postbiotics, therefore I present only the knowledges published so far. Future head-to-head 
studies will be necessary to determine the best parent cell strains, the best postbiotic dosages, and the 
inexpensive postbiotics in comparison to other medications Postbiotics are viable therapy option for 
human patients, paving the path for the development of new pharmaceutical and food items with 
specialized physiological effects. The use of postbiotic chemicals in the preservation of meat products 
and dairy products could be a novel strategy. Exploring the human internal bioreactor may be a light of 
hope in the midst of heavy clouds; consequently, a vision of “precision postbiotics” will undoubtedly 
emerge. Further research is expected into bioactivities of these metabolites which will unveil potential 
uses for postbiotics in medicine.
6. Ensure consistency in the nomenclature used for bacterial strains and components throughout the 
text. For example, use the same naming convention for strains of Lactobacillus (e.g., L. rhamnosus) 
and ensure consistency in abbreviations (e.g., TNF-α vs. TNF-alpha).
I apologize for the mentioned mistakes in the text, the names have been edited
7. Explore molecular mechanisms underlying the observed immunomodulatory effects of cell wall 
components, such as signaling pathways involved in cytokine production or interactions with host 
receptors.
The major components of the cell wall are peptidoglycan and teichoic acid which have 
immunomodulatory effects. Most of the study showed that peptidoglycan produced by Lactobacillus is 
responsible for inhibiting the pro-inflammatory cytokine production. Peptidoglycan is a characteristic 
cell wall component of Gram-positive bacteria and can stimulate macrophages and dendritic cells to 
secrete proinflammatory cytokines. The one study revealed that exist two types of lactobacilli (sensitive 
to intracellular digestion by macrophages and resistant to intracellular digestion). The sensitive 
Lactobacillus strains inhibit the IL-12 production and Lactobacillus resistant to intracellular disgestion 
by macrophage can effective stimulate secretion of IL-12. This novel observation suggests that 
peptidoglycan can act in both proinflammatory and anti-inflammatory ways. Dysregulated 
overproduction of IL-12 and the subsequent overactivation of T helper (Th)1 cells are considered to be 
one of the causes of autoimmune diseases and inflammatory bowel diseases.
The first identified peptidoglycan receptors were the nucleotide-binding oligomerization domain 1 and 2 
(NOD1 and NOD2). NOD1 and NOD2 are cytosolic receptors that directly detect fragments of 
peptidoglycan shed from bacteria or generated during the degradation of bacteria in the phagolysosome.
Peptidoglycan-induced iNOS, COX-2 and proinflammatory cytokine expression is mediated through the 
TLR2/MyD88/PI3-kinase/AKT pathway, which in turn initiates IKK alpha/beta and NF-kappaB activation.
8. In the Introduction section the author should refer to the research paper and comment on recent insilico techniques. It will be good information for the readers. I would like to recommend several papers, 
among many others, providing further explanation on this topic:PMID: 33749525 PMID: 23564489 
PMID: 23852834
I thank the reviewer for the aforementioned research works and in-silico techniques, in silico methods -
theoretical methods that include computer models that serve to predict the effects of substances (return 
of postbiotics). This estimate of the effect on the organism needs to be supplemented with in vivo and 
in vitro methods. They are specific methods that can be used in the future.
9. The author should clarify the specific metabolic disorders and molecular pathways targeted by 
postbiotics to provide a more focused understanding of their therapeutic potential.
Postbiotics, including short-chain fatty acids (SCFAs), vitamins, exopolysaccharides, enzymes, cell-free 
supernatants, and bacteriocins have a crucial role in addressing metabolic diseases by exhibiting 
diverse biological activities. They exert antiobesity effects by improving P38 AMP-activated protein 
kinase (AMPK) phosphorylation and enhance anti-inflammatory effects by inhibiting cyclo-oxygenase-2 
(COX-2), IL-6, IL-12, and TNF-α, along with acceleration of immune-modulation activity by inhibition of 
extracellular signal-regulated kinase (ERK), c-Jun N-terminal kinase (JNK), and p38 mitogen-activated 
protein kinase (MAPK) phosphorylation. 
Postbiotics also exert antihyperglycemic effects, improving insulin sensitivity and regulating glucose 
metabolism, particularly relevant for diabetes, via interferon regulatory factor 4 (IRF 4) and nucleotidebinding oligomerization domain 2 (NOD-2) intervention. Given their antimicrobial properties, postbiotics 
help maintain a healthy gut microbiota, preventing dysbiosis. Additionally, their antioxidant activity 
protects cells from oxidative stress by inhibiting reactive oxygen species (ROS) and reactive nitrogen 
species (RNS) scavenging properties. 
10. The author should provide insights into the mechanisms underlying the observed bioactivities of 
postbiotics, including their effects on specific cellular pathways and physiological processes.
The composition of the intestinal microbiota has the potential to influence physiological processes in the 
organism and if the composition of intestinal microbiota is changed by inappropriate diet, sedentary 
lifestyle and lack of physical activity, processes lead to pathogenesis of different disorders
There are numerous ways to enhance the population and diversity of intestinal microbiota for example: 
eating fermented food, consumption of lots of vegetables, having whole grain foods, consuming foods 
rich in polyphenols, maintaining a plant-based diet, having prebiotics or probiotics, etc. Another 
strategical measure to modulate the gut microbiota has introduced the new member of biotics family, 
postbiotics, that directly or indirectly exert valuable benefit on the host’s body (as mentioned in point 3).
11. The author should discuss the potential synergistic effects of combining postbiotics with probiotics 
or prebiotics to enhance their therapeutic efficacy and broaden their applications.
The topic of combining postbiotics with probiotics and prebiotics would be suitable for a completely new 
manuscript. It is important to clarify that probiotics and prebiotics form together synbiotics, that means 
that prebiotics potentiate the effect of probiotics, they are food for only probiotics. Their metabolism 
produces metabolites that represent postbiotics, and some of their mechanisms are described in the 
manuscript. Probiotics multiply and expand in the intestine, but postbiotics do not, and as a result, do 
not have a sustained effect if treatment is stopped.
12. The author should consider the implications of postbiotic research for personalized medicine 
approaches, including identifying biomarkers for predicting individual responses to postbiotic 
interventions.
13. The author should discuss the potential impact of postbiotics on public health and healthcare 
systems, including their role in preventing and managing chronic diseases and reducing healthcare 
costs.
14. The author should discuss the potential environmental and sustainability implications of scaling up 
postbiotic production and consumption, including resource use and waste management considerations.
Answer for ponts 12,13,14
The manuscript is entitled "biotherapeutic potential", the described mechanisms of action of postbiotics 
are aimed at preventing health in general, at individual diseases, including their risk factors that cause 
them. Diseases are a big burden, they are associated with enormous health care, their production and 
applicability mainly in the form of a personalized approach will reduce the costs of health care, but mainly 
improve the quality of life for individuals.
I am a researcher, I do not deal with the issue of the consequences of increasing postbiotic production 
and consumption, nor with waste management, so I do not comment on the topic.
15. Clarify the definition of postbiotics for readers who may not be familiar with the term.
The exact definition of postbiotics is debated, as I mentioned in the manuscript. I would explain this term 
as I simply stated it in answer point 1 and 11 .
15. Expand on the potential mechanisms through which postbiotics could exert their effects on 
metabolic disorders.
The most studied postbiotics, SCFAs, metabolites derived from intestinal bacteria are known modulators 
of host metabolism, have been suggested as potential modulating and /or preventive factors of 
cardiovascular diseases, including type 2 diabetes mellitus and obesity.
SCFAs exert benefits in various diseases which are characterized by a deregulation of the blood 
pressure, glucose, and lipid metabolism, inflammation response, and gut integrity.
SCFA can act using two different mechanisms: 1) direct action on the enterocytes, maintaining the gut 
barrier integrity – fuel enterocytes, increase production of mucin, increase tight junction proteins and 
antimicrobial peptide and decrease LPS and pH or 2) indirect action of SCFA regulate the inflammatory 
and immune response (increase T-cells, and anti-inflammatory ILs, and de-crease pro-inflammatory 
cytokines), blood pressure (binding to GPR-41), increases satiety, and maintenance lipid and glucose 
homeostasis (increasing insulin sensitivity, fatty acid oxidation and decreasing plasma level of glucose, 
lipid accumulation, gluconeogenesis and plasma cholesterol), through the mechanisms:
a) Inhibition of K/HDAC (lysine/histone deacetylase) leads to histone hyperacetylation, which turns in a 
higher accessibility of transcription factors to the promoter regions of different genes;
b) Activation of signaling transduction by AMPK (AMP-activated protein kinase) and NFκB in the small 
intestine, colon, liver, spleen, heart, skeletal muscle, neurons, im-mune cells, and adipose tissues, and 
GLP-1 and PYY secretion in intestinal enteroendocrine L-cells caused by the binding of SCFAs to the 
GPR, increase of cAMP (cyclic adenosine monophosphate) levels by the binding of propionate or 
acetate to the receptor Olfr78/OR51E2 (in vascular smooth muscle cells in the peripheral vasculature 
and renal afferent arteriole). GLP-1 and PYY enter into the systematic circulation exerting benefit in 
different tissues and cells;
c, Butyrate works as a ligand of the AHR (aryl hydrocarbon receptor) and PPARy (peroxisome 
proliferator-activated receptor), leading to the expression of genes dependent on these transcription 
factors.
16. Provide more specific examples of natural compounds obtained from microorganisms that have 
shown promise as postbiotics.
Research and development in the field of postbiotics continues primarily in the food industry and the 
pharmaceutical industry, the development of new techniques will certainly contribute to the identification 
of new postbiotics, therefore I present only the knowledges published so far.
17. Discuss any existing research on the safety and efficacy of postbiotics in humans, particularly in 
the context of metabolic disorders.
Probiotics multiply and expand in the intestine, but postbiotics do not, and as a result, do not have 
a sustained effect if treatment is stopped. Postbiotics possess favorable influence on the host, are 
compounds and/or cell wall constituents produced by living bacteria or liberated after bacterial lysis. 
Even if the precise mechanisms are still unclear, postbiotics may help the health of host by imparting 
certain physiological advantages. To enable the discovery and characterization of newer postbiotics, 
which can help with the comprehension of signaling pathway alteration, more research is still required. 
Postbiotics, whether as bacteria’s components or metabolites, offer benefits and can mimic probiotics’ 
favorable therapeutic effects. They do so without putting the body at risk of receiving live germs.
Postbiotics aid in bio preservation and the reduction of biofilm development in food due to their organic 
acids, bacteriocins, and other antibacterial activities.
Postbiotics are include in the active packing system „ antimicrobial active packing“ where protect food 
from microbial decomposition during transportation and storage as antimicrobial agents Postbiotics are 
potential options since any drugs that hve the ability to improve the intestinal barrier can be recognized 
as health-promoters.
18. Consider including information on the potential side effects or limitations associated with the use of
postbiotics.
The use of postbiotics as therapeutics agents faces several obstacles that must be addressed before 
they can be successfuly implemented in clinical practise. As I mentioned the use of postbiotic chemicals 
in the preservation of meat products and dairy products could be a novel strategy, therefore is question 
what chemical? One impostant issue is a lack of standarized postiotics definitions and criteria, which 
makes it difficult to compare research and products. Optimizing of dose is difficult because the optimal 
dose is determined by factors such as the specific postbiotic, the desired disease, and the patient´s 
charcteristocs. Distribution technique must be carefully selcted, takineg into account characteristics 
such stability, bioavailability, and targeted distribution to specific areas of gastrointestinal tract. 
Understanding the mechansm of action, the specificity of effects, and integartion with conventional 
therapy are also ongoing problem. Despite these obstacles, continos research and development efforts 
can pave the way for the successful use of postbiotics in therapeutics applications.
19. Consider including a section on the current methods used to identify and characterize new 
postbiotics.
„Criteria for a preparation to qualify as a postbiotics include: molecular characterization of the progenitor 
microorganisms to allow accurate identification and screening of potential genes of safety concern, a 
detailed description of the inactivation procedure and confirmation that inactivation has occurred, a 
detailed description of the composition of the postbiotics preparation and assessment of the safety of 
the postbiotics in the target host and evidence of health benefit in the host from a controlled, high-quality 
studies“
New methods used to identify and characterize new postbiotics will bring further research based on 
animal models or clinical studies

Reviewer 2 Report

Comments and Suggestions for Authors

The review emphasizes the potential significance of postbiotics as a promising therapeutic avenue. It underscores the importance of exploring natural compounds derived from microorganisms for their effects on metabolic disorders, molecular pathways, immune system modulation, and intestinal microbiota in both children and adults. Author aims to contribute insights that can guide future research in identifying novel postbiotics and understanding additional mechanisms to address metabolic diseases. . Overall, the manuscript demonstrates good writing, and effective execution. However, some areas could be improved to enhance the quality and impact of the manuscript.

The author can include sections on the Metabolic Pathways of Postbiotics, covering aspects such as the generation of Postbiotic Metabolites, the biochemical attributes of these compounds, and their influence on modulating the gut microbiota. A dedicated section discussing the Biotherapeutic Potential of Postbiotics should also be incorporated.

Author Response

Cover letter for reviewer 2
Thank you for the review and I attach explanations. 
The review emphasizes the potential significance of postbiotics as a promising therapeutic avenue. It 
underscores the importance of exploring natural compounds derived from microorganisms for their 
effects on metabolic disorders, molecular pathways, immune system modulation, and intestinal 
microbiota in both children and adults. Author aims to contribute insights that can guide future research 
in identifying novel postbiotics and understanding additional mechanisms to address metabolic 
diseases. . Overall, the manuscript demonstrates good writing, and effective execution. However, some 
areas could be improved to enhance the quality and impact of the manuscript.
The author can include sections on the Metabolic Pathways of Postbiotics, covering aspects such as 
the generation of Postbiotic Metabolites, the biochemical attributes of these compounds, and their 
influence on modulating the gut microbiota. A dedicated section discussing the Biotherapeutic Potential 
of Postbiotics should also be incorporated.
Postbiotics possess favorable influence on the host, are compounds and/or cell wall constituents 
produced by living bacteria or liberated after bacterial lysis. I do not mention the generation of postbiotics 
in the manuscript, because several authors decribe the generation of postbiotic. Their effect on the 
modulation of intestinal microbiota is indicated individually for individual postbiotics if their effect was 
recorded.
Postbiotics maintan host health and many act as a mediator for the positive effects via a number of 
pathways, but the targeted mode of action is still being studied, therefore I am not including metabolic 
pathways in the manuscript except the mechanism of the action of SCFA, the most studied postbiotics 
The most studied postbiotics, SCFAs, metabolites derived from intestinal bacteria are known modulators 
of host metabolism, have been suggested as potential modulating and /or preventive factors of 
cardiovascular diseases, including type 2 diabetes mellitus and obesity, help in regulation of blood 
pressure, glucose, and lipid metabolism, inflammation response, and gut barrier integrity.

Reviewer 3 Report

Comments and Suggestions for Authors

The present review article focuses on postbiotics, which include bacteria, microorganism-derived cell components, and metabolites. These postbiotics provide potential benefits over probiotics and are considered safer for usage in human beings. This thorough selection seems to be both intriguing and practical. However, the following issues will be addressed.

The definition of postbiotics should be provided with comprehensive elaboration. Dysbiosis plays a crucial role in maintaining the balance of gut microbiota. The alteration of helpful or hazardous organisms during the postbiotics period seems to be overlooked.

A particle has been included in each segment. Nevertheless, it is possible to compare postbiotics for both children and adults by presenting the information in a single table that clearly indicates the specific microbiota for each.

The development of metabolic diseases is linked to the makeup of the gut microbiota. Nevertheless, the postbiotics were not included in this section.

The conclusion did not include the example of postbiotics.

The present report may be strengthened by its limitations.

Comments on the Quality of English Language

The language seems better to check through a professional editing service.

Author Response

Cover letter for reviewer 3
Thank you for the review and I attach explanations. 
The present review article focuses on postbiotics, which include bacteria, microorganism-derived cell 
components, and metabolites. These postbiotics provide potential benefits over probiotics and are 
considered safer for usage in human beings. This thorough selection seems to be both intriguing and 
practical. However, the following issues will be addressed.
The definition of postbiotics should be provided with comprehensive elaboration. Dysbiosis plays a 
crucial role in maintaining the balance of gut microbiota. The alteration of helpful or hazardous 
organisms during the postbiotics period seems to be overlooked.
A particle has been included in each segment. Nevertheless, it is possible to compare postbiotics for 
both children and adults by presenting the information in a single table that clearly indicates the 
specific microbiota for each.
The development of metabolic diseases is linked to the makeup of the gut microbiota. Nevertheless, 
the postbiotics were not included in this section.
The conclusion did not include the example of postbiotics.
The present report may be strengthened by its limitations.
Postbiotics maintan host health and many act as a mediator for the positive effects via a number of 
pathways, but the targeted mode of action is still being studied,
As I write in manuscript „Although the exact definition of postbiotics is debated, postbiotics are defined 
as non-living microorganisms without losing their original function and providing a health benefit to the 
host [11-13]. Common types of postbiotics, known as metabiotics are paraprobiotics or “ghost probiotics” 
and fermented infant formula“ there is no exact definition, so we can also define 
postbiotics as „non-viable bacterial products or metabolic products from microorganisms that have 
biological activity in the host“, or 
postbiotics are metabolites and cell-wall components that are beneficial to the host and are released 
by living bacteria or after lysis,
parabiotics as “inactivated microbial cells (non-viable) that confer a health benefit to 
the consumer” and 
metabiotics as “ the structural components of probiotic microorganisms and/or their metabolites and/or 
signaling molecules with a determined (known) chemical structure that can optimize host-specific 
physiological functions“.
Comparison of the application of postbiotics for children and adults are not drawn in simple tables, but 
are presented in individual parts of the manuscript.
The composition of the intestinal microbiota has the potential to influence physiological processes in the 
organism and if the composition of intestinal microbiota is changed by inappropriate diet, sedentary 
lifestyle and lack of physical activity, processes lead to pathogenesis of different disorders including 
metabolic disorders. The results of use of individual postbiotics is included in part 4 and part 5 and some 
of them have shown changes in the intestinal microbiota documented in the manuscript. 
The examples of postbiotics are included in all manuscript.
I do not what means"postbiotic period", beacuse probiotics multiply and expand in the intestine, but 
postbiotics do not, and as a result, do not have a sustained effect if treatment is stopped.
